# Impact of Thermal Dissipation on the Lighting Performance and Useful Life of LED Luminaires Applied to Urban Lighting: A Case Study

**DOI:** 10.3390/ijerph19020752

**Published:** 2022-01-10

**Authors:** Juan de Dios Unión-Sánchez, Manuel Jesús Hermoso-Orzáez, Manuel Jesús Hervás-Pulido, Blas Ogáyar-Fernández

**Affiliations:** 1Centre for Advanced Studies in Energy and Environment, University of Jaén, 23071 Jaén, Spain; junion@ujaen.es (J.d.D.U.-S.); bogayar@ujaen.es (B.O.-F.); 2Department of Graphic Engineering, Design and Projects, University of Jaén, 23071 Jaén, Spain; mjhp0001@red.ujaen.es; 3Department of Electrical Engineering, University of Jaén, 23071 Jaén, Spain

**Keywords:** LED, thermal dissipation, luminaire, CFD (computational fluid dynamics), FMV (finite volume method)

## Abstract

Currently, LED technology is an established form of lighting in our cities and homes. Its lighting performance, durability, energy efficiency and light, together with the economic savings that its use implies, are displacing other classic forms of lighting. However, some problems associated with the durability of the equipment related to the problems of thermal dissipation and high temperature have begun to be detected, which end up affecting their luminous intensity and the useful life. There are many studies that show a direct relationship between the low quality of LED lighting and the aging of the equipment or its overheating, observing the depreciation of the intensity of the light and the visual chromaticity performance that can affect the health of users by altering circadian rhythms. On the other hand, the shortened useful life of the luminaires due to thermal stress has a direct impact on the LCA (Life Cycle Analysis) and its environmental impact, which indirectly affects human health. The purpose of this article is to compare the results previously obtained, at different contour temperatures, by theoretical thermal simulation of the 3D model of LED street lighting luminaires through the ANSYS Fluent simulation software. Contrasting these results with the practical results obtained with a thermal imaging camera, the study shows how the phenomenon of thermal dissipation plays a fundamental role in the lighting performance of LED technology. The parameter studied in this work is junction temperature (T_j_), and how it can be used to predict the luminous properties in the design phase of luminaires in order to increase their useful life.

## 1. Introduction

Currently, energy saving in public lighting is generating great interest and has become a priority in the management of outdoor lighting in our cities; for this reason, LED luminaires are becoming more and more frequent [1]. There are studies related to the thermal dissipation of LEDs applied to automotive lighting that systematically prove that the properties of thermal interface materials, such as thermal resistance, affect the optical properties of the luminaire [2]. Other studies demonstrate the loss of chromaticity and degradation of light, the change in chromaticity, and transmittance loss tested in phosphor converted white light emitting diodes (PC-WLED) under accelerated thermal tests at 150 °C, 200 °C, and 250 °C [3]. On the other hand, the increasing use of LED lights has modified the natural light environment dramatically, posing novel challenges to both humans and wildlife. Indeed, several biomedical studies have linked artificial light at night to the disruption of circadian rhythms [4], with important consequences for human health, such as the increasing occurrence of metabolic syndromes [5], cancer [6], and reduced immunity [7]. In this sense, there are many studies that clearly show the adverse effects of poor lighting on health, including light depreciation as a consequence of deterioration, faulty thermal dissipation, or the devaluation of the light intensity and chromaticity of the luminaire [8,9].

The chromaticity, electrical properties and thermal properties of LED devices are highly dependent on each other [10]; therefore, heat dissipation plays an important role in improving the efficiency and reliability of LED lighting [11]. Deficiencies in artificial lighting, loss of light intensity, or changes in chromaticity can have serious consequences on human health in relation to circadian rhythms [12,13]. This is why it is extremely important to install energy-efficient outdoor lighting equipment, but that have, at the same time, a low impact on human health [14,15]. Shorter wavelengths of light preferentially disturb melatonin secretion and cause circadian phase shifts, even if the light is not bright [13].

The main source of heat in an LED comes from the union between the p-type and the n-type semiconductor material that makes up the device [16]. The LEDs disperse 45% of the energy applied in light and the remaining 55% in heat. High-power chips increasingly emit more heat, which leads to performance degradation [17] and, ultimately, results in a shorter life expectancy of LED products. Therefore, the industry is investigating product design structures to control the heat generated by high power LED chips [18].

The temperature reached at the p–n junction is called junction temperature (T_j_) and is considered a key parameter in LED performance [19]. The junction temperature at which light-emitting diodes have to operate must be low, as there is an inverse relationship between junction temperature and the lifetime of the LED [20].

The internal thermal resistance and junction temperature (T_j_) are the critical parameters of LEDs, which must be maintained at nominal levels for reliable and robust operation [21].

For a given set of operating conditions, an engineer can calculate the junction temperature of an LED [22] and design a thermal dissipation system to lower the temperature of electronic devices and extend the life of LEDs [23].

Modern chips are designed with conductive heat pipes to channel heat from the junction to the “solder point”. The welding point is the part of the LED that comes in contact with the PCB (printed circuit board) and/or independent heat sink.

The object of this study is to propose an analysis methodology to calculate the theoretical thermal dissipation, a priori, depending on the design, the materials and the operating ambient temperature of the LED luminaires to later compare and discuss the theoretical results with the experimental ones made with a thermographic camera. The comparison of results between the simulations and the real dissipation in the laboratory allow to analyze the junction temperature of the LEDs because it is a parameter that cannot be measured through a thermographic camera. To do this, we thermally analyze, according to the proposed method, a LED luminaire, carrying out simulations, at different working environment temperatures [24]. Considering that thermal dissipation is essential to maintain the luminous properties of LED luminaires, directly impacting the useful life of the equipment and its light quality [25], in direct relation to the impact that these deficiencies can have on the circadian rhythms of people exposed to the light artificial defective [26,27]. Likewise, the useful life of LED luminaires is related to the quality of the equipment and the ability of the materials to dissipate radiated heat, this aspect directly impacting the LCA (life cycle analysis) on the carbon footprint [28] and environmental impacts of the manufacture of new substitute equipment [25,29]. Today, it is very important to invest on sustainable luminaires with a low carbon footprint and high durability [28].

## 2. Methodology

The methodology applied in this work has two parts. First, a simulation of the thermal dissipation of a LED luminaire model applied to urban lighting under different boundary conditions was carried out.

The Model luminaire corresponds to a high-power LED luminaire with a specific design to achieve an adequate lighting performance and optimal thermal dissipation. We proceeded to analyze the results and observe the theoretical heat dissipation that occurs in it using the ANSYS Fluent software [30].

Later, this work is complemented with a validation analysis of the methodology with the practical results realized with a thermographic camera. We analyzed the actual results of the thermal dissipation of the luminaire with respect to the boundary conditions in the lighting laboratory [31], trying to assess and contrast the theoretical results, expected a priori in the design phase, with the real and practical results obtained [32]. We will try to justify that an adequate design and the choice of materials are key elements that allow a better heat dissipation in LED luminaires, improving their stability and lighting performance [33].

The analysis through CFD (computational fluid dynamics) simulation is represented in Figure 1, through which we obtained the theoretical results of the thermal dissipation that were produced by the Model and thus be able to make the comparison between the data obtained between the theoretical simulations and the experimental data obtained with the thermal imaging camera [34].

The computational dynamics of fluids is the science that predicts the flow of a fluid, the transfer of heat and mass, chemical reactions, and related phenomena through the numerical resolution of the set of mathematical equations.

CFD analysis complements testing and experimentation by reducing the total effort and cost required for experimentation and data acquisition.

In the numerical simulation, there are three stages: pre-processed, processed and post-processed.

### 2.1. Pre-Processing

#### 2.1.1. Select the Luminaires for the Study

The study was carried out by selecting a high-power luminaire manufactured with a novel design, specially oriented to the improvement of the thermal dissipation, and is referred to as Model.

#### 2.1.2. Obtaining the Information on the Luminaire to Be Analyzed

The manufacturer provided the different characteristics and properties of the luminaire, who remains confidential in this project.

The Model consists of different components and materials that are attached in Table 1. The total nominal power of the luminaire is 204 W, of which 192 W is for the operation of the LEDs and 12 W for the drivers.

The drivers are composed of an electronic circuit that performs the following functions:-To transform alternating current (AC) into direct current (DC), which is used by LEDs for their correct operation;-To adapt the output voltage and current to the LED requirements.

#### 2.1.3. Elaboration of the 3D Model of the Equipment

The elaboration of the model was carried out using CATIA (Computer-Aided Three-Dimensional Interactive Application) [35], which consists of computer-aided design, manufacturing and engineering software (Figure 2). 

#### 2.1.4. Discretization of the 3D Model for Study

To execute the discretization, the specific program Altair HyperMesh was used. It is a computer-aided engineering simulation software (CAE, computer aided engineering) platform, in which it is possible to create finite element models for the analysis and prepare meshes of high quality efficiently. The method of discretization in the 3D model was in finite volumes [36].

A two-dimensional mesh was made with triangular elements (triads). Having a control of the mesh with skew means that an angle between the vector from each node to the midpoint of the opposite side and the vector between the two adjacent middle sides in each node of the element is 90 degrees minus the angle found using skew. In this study, the skew sought is 60 degrees, providing a triangle as similar as possible to an equilateral triangle for the subsequent generation of 3D elements. See Figure 3 and Figure 4.

##### Luminaire Discretization Model

Next, in the series of Figure 5, Figure 6 and Figure 7, we represent the discretized components.

The mesh was generated in triads and a tetrahedral, although with square meshes and hexahedrons can be provided higher quality solutions with fewer cells/nodes. The square and hexahedral meshes show a reduced numerical diffusion when the mesh is aligned with the flow, but more effort is also required in generating a square mesh and hexahedrons.

#### 2.1.5. Configuration of the Physical and Solver Characteristics

##### Definition of the Properties of Materials

It is necessary to define the properties of the materials that are going to be assigned to the different components of the luminaire to be taken into account during the study and thus have greater precision in the results when compared with the experimental data, reflected in Table 2.

##### Definition of Physical Models

Conservation equations.

The ANSYS Fluent solver is based on the finite volume method [37]. The domain is discretized in a finite set of control volumes and the general conservation equations for mass, momentum, energy, etc., are solved in this set of control volumes. All the equations are then solved numerically to represent the solution field.

2.Finite Volume Method.

The general equations of conservation of the mass, amount of movement, energy, etc. are resolved in this set of control volumes. In the centroid of each control volume, there is a node where the value of the variables is calculated and, in the borders, their value can be known through interpolation. Then, we continued with an approximation of the conservation equations to determine a system of algebraic equations and obtain their solution by iterative methods.

Starting from the general transport equation, obeying the Navier–Stokes equations in the integral form [38]:(1)$$\frac{d}{\mathrm{dt}}\int_{V}^{\;}\rho \Phi \mathrm{dV}+\oint_{A}^{\;}\rho \vec{v}\Phi \cdot \vec{n}\mathrm{dA}=\oint_{A}^{\;}\rho \Gamma \nabla \Phi \cdot \vec{n}\mathrm{dA}+\int_{V}^{\;}S_{\phi }\mathrm{dV},$$ where Φ = variable transported by a medium;ρ = density of the medium through which it is transported Φ;V = travel speed of Φ through the medium;Γ = medium diffusion constant;$S_{\phi }$ = source/sink term of the variable transported;A = border;$\vec{v}$ = speed vector of Φ through the medium;$\vec{n}$ = normal vector to the surface.


The first term corresponds to the temporal variation of the variable transported by the medium within a volume; the second term corresponds to the convective flows of the variable transported across the border due to the speed. The first term after equality expresses the diffusive term of the variable transported at the border that depends only on the gradient of said variable, and the last term is due to the source term of the variable inside the volume.

The source term, $S_{\phi }$ of Equation (1), contains the radiation terms calculated through the discrete ordinate model (DO). The DO model transforms the radioactive transport equation for an absorption, emission and dissipation medium in the position $\vec{r}$ and in the direction $\vec{s}$ in a transport equation for the radiation intensity in the spatial coordinates (x, y, z). The resolution method is the same as that used for the flow and energy equations, and represented in Equation (2) [38]:(2)$$\frac{\mathrm{dI}\left(\vec{r},\vec{s}\right)}{\mathrm{ds}}+\left(a+\sigma_{s}\right)I\left(\vec{r},\vec{s}\right)={\mathrm{an}}^{2}\frac{{\sigma T}^{4}}{\pi }+\frac{\sigma_{s}}{4\pi }\int_{0}^{4\pi }I\left(\vec{r},{\vec{s}}^{\prime }\right)\Phi \left(\vec{s}\cdot {\vec{s}}^{\prime }\right)d\Omega^{\prime }$$
where $\vec{r}$ = position vector;$\vec{s}$ = direction vector;${\vec{s}}^{\prime }$ = vector direction dissipation;s = path length;a = absorption coefficient;n = refractive index;$\sigma_{s}$ = dispersion coefficient;σ= Constant Stefan–Boltzmann (5669 × 10^−8^ W/m^2^ K^4^);I = Intensity of radiation;T = local temperature in Kelvin;Φ = phase function;$\Omega^{\prime }$ = solid angle.


For finite volumes and sub-border volumes that correspond to the volume boundary, we obtained [38]:(3)$$V_{\mathrm{cell}}\cdot \frac{\partial \Phi }{\partial t}+\sum_{f}^{\mathrm{Nfaces}}\rho_{f}{\vec{v}}_{f}\Phi_{f}\cdot {\vec{A}}_{f}=\sum_{f}^{\mathrm{Nfaces}}\Gamma_{\phi }\nabla \Phi_{f}\cdot {\vec{A}}_{f}+S_{\phi }+V_{\mathrm{cell}}\;$$
where N_faces_ = number of faces of volume;f = fluid.


The differential equations were discretized in a system of algebraic equations that are solved numerically to give a field of solutions.

Implement boundary conditions

The boundary conditions were determined to study the behavior of the luminaire in certain cases. The boundary conditions were determined in the air, which is found surrounding the luminaire. Several environmental temperatures have been attempted to be simulated [24], such as, 40 °C, 20 °C and −10 °C, with an initial speed ascending of 0.02 m/s.

3.Determination of the problem solver.

ANSYS Fluent allows one to choose one of the two numerical methods:-Pressure-based solver;-Density-based solution.

In both methods, the velocity field is obtained from the moment equations. In the density-based approach, the continuity equation is used to obtain the density field, while the pressure field is determined from the state equation.

On the other hand, in the pressure-based approach, the pressure field is extracted by solving a pressure or pressure correction equation that is obtained by manipulating the continuity and momentum equations.

In our case, we used the pressure-based solver, which uses an algorithm that belongs to a general class of methods, called the projection method.

There are two pressure-based solution algorithms available in ANSYS Fluent. A segregated algorithm and a coupled algorithm.

-Segregated algorithm based on pressure: The individual governing equations for the solution variables (for example, u, v, w, p, T, k, etc.) are solved one after the other. The convergence of the solution is relatively slow, since the equations are resolved in an uncoupled way;-Algorithm coupled based on pressure: The pressure-based coupled algorithm solves a coupled system of equations that comprises the moment equations and the pressure-based continuity equation. Since the equations of momentum and continuity are solved in a tightly coupled manner, the convergence rate of the solution improves significantly compared to the segregated algorithm. The convergence with this algorithm improves with respect to the segregated algorithm, which is taken into account in the choosing of this method.

4.Monitorization of residuals.

In a CFD analysis, the residue measures the local imbalance of a variable stored in each control volume. For CFD, the residual levels of 10^−4^ are considered to be slightly convergent, the levels of 10^−5^ are considered to be very convergent, and the levels of 10^−6^ are considered to be closely convergent.

### 2.2. Processed

The residual definitions that are useful for a problem class are sometimes deceptive for other kinds of problems. Therefore, it is a good idea to judge convergence not only by examining residual levels, but also by monitoring relevant integrated quantities, such as the heat transfer or drag coefficient.

In the present analysis, continuity, velocity in x, velocity in y, velocity in z, energy and DO with a convergence criterion 10^−3^, 10^−3^, 10^−3^, 10^−3^, 10^−6^ and 10^−5^ were taken into account, respectively.

### 2.3. Post-Processing

The results were examined to review the solution and extract the data. The tool used is Altair HyperView, a complete post-processing and visualization environment for CFD. It allows to visualize the data in an interactive way.

## 3. Results

The results section is divided into two parts. On the one hand, the theoretical results show the data obtained with the simulation software at the three boundary temperatures, as well as the validation of the luminaire materials by comparing the limit temperatures of each of the materials that make up the luminaires with the operating temperatures obtained in the simulation for each of them.

Finally, in the practical results, the thermal dissipation of the LED luminaire is tested at an ambient temperature of 20 °C in order to validate the theoretical results obtained previously.

### 3.1. Theoretical Results

#### 3.1.1. Display of Temperature and Air Speed in the Model Luminaire at 20 °C

In the next Figure 8 and Figure 9, we graphically display in color the temperature of different parts of the Model at 20 °C.

#### 3.1.2. Display of Temperature and Air Speed in the Model Luminaire at 40 °C

In the next Figure 10 and Figure 11, we graphically display in color the temperature of different parts of the Model at 40 °C.

#### 3.1.3. Display of Temperature and Air Speed in the Model Luminaire at −10 °C

In the next Figure 12 and Figure 13, we graphically display in color the temperature of different parts of the Model at −10 °C.

In the theoretical data obtained from the simulations, a validation of the materials is carried out followed by an analysis of the LEDs. The validation of the materials consists of the comparison of the limit temperature that the material used for its operation can reach without having problems of deterioration with the maximum temperature reached in each component of the luminaire to observe if the limit temperature of the material is exceeded.

Regarding the drivers, they are not composed of only one material, as it is a combination of several subcomponents. The weakest components to the temperature are the microchips that are assembled on the driver PCB, then in the comparison we have divided the driver information in two rows (bases and electronics components) and for electronics we select the temperature limit of the microchips for the evaluation of the driver (Table 3, Table 4 and Table 5).

In the next Figure 14, Figure 15 and Figure 16, a comparison of some of the different components of the Model and for the different ambient temperatures of the simulations is shown, giving the range of temperatures at which, the components are located.

The heatsink is made of aluminum and has a limit temperature of 460 °C. The maximum temperature reached by the heatsink in the simulations is much lower than the limit temperature. The diffuser is composed of PC (polycarbonate), a material that has an operating limit temperature of 145 °C. The results of the simulations show that the maximum temperatures of these components are within operating temperature range, as shown in Figure 17.

### 3.2. Experimental Results

The prototype high-power LED luminaire was built with the same geometry and dimensions than the model in order to validate the simulation. This luminaire corresponds to Air Series 7 of ATP lighting (Figure 18).

In the following Table 6 and Table 7, the main power characteristics of LED and driver features are collected.

The drivers used for control LEDs is the chip MP4688 high power control LEDs. The luminaire has two drivers PCB, and each driver PCB has three MP4688. Each integrated MP4688 controls 16 LEDs (96 LEDs in the luminaire).

Each MP4688 has nominal power of 2W. As we have a nominal power of 2 W, two drivers PCB and three chips in each PCB, we have a total value of 12 W (total power drivers).

For the acquisition of real thermal dissipation data of the analyzed luminaire, a thermographic camera was used at a stabilized temperature of 20 °C and in a climatized environment with a relative humidity of 70% and air speed of 0 m/s.

The thermal imaging camera used was the FLUKE Ti 25 model. It captures a digital image with an infrared and fuses it. The temperature range is −20 °C to 350 °C with a precision of ±2 °C. The thermal sensitivity is 0.1 °C at an ambient temperature of 20 °C, which describes the smallest difference between two pixels in temperature that the camera can measure (Table 8).

A fundamental parameter when acquiring images with the thermal imaging camera is the emissivity. All objects radiate infrared energy. The amount of energy radiated is based on two main factors: the surface temperature of the object and the emissivity of said surface. Emissivity is a very important issue for measuring surface temperature without being in contact [39]. Depending on the material to be measured, the emissivity varies [40]. The emissivity determined to obtain the temperatures of the materials was obtained from a table of emissivities.

A progressive data collection was made from the ignition to the thermal stabilization of the equipment, taking data periodically until the temperature remained stable.

To take the measurements, the dissipater and diffuser with the thermal imager and to be a specific material with a specific alloy or formulation in the material compound, several types of emissivities were used to obtain the exact temperature (Figure 19, Figure 20, Figure 21 and Figure 22).

## 4. Discussion

Once the simulation was validated at 20 °C in the laboratory, an analysis of the junction temperature of the LEDs (T_j_) is carried out in order to study how it affects the lighting performance.

To ensure the useful life, efficiency and color of the LEDs, the junction temperature of the LEDs must be maintained in a specific range.

After obtaining the junction temperature of the LEDs in the simulations for the different boundary conditions that surround the luminaire, such as 40 °C, 20 °C and −10 °C, they are compared with the LED junction temperatures found in the market (Table 9, Figure 23 and Figure 24).

For the Model, an Osram LUW CQAR LED (streetwhite) with its respective data sheet was used. It was chosen to take note of the dimensions of the LED with the power of 2 W/LED and, therefore, have an orientation for the features. The maximum junction temperature of this LED corresponds to a temperature of 135 °C. In this luminaire, the maximum temperature reached is 10 °C above the value of the temperature of the LED junction, which causes a decrease in lamp life, possibly causing the LED light output to decrease irreversibly in the long term at a faster rate than at lower temperatures [41]. Controlling the temperature of LEDs is, therefore, one of the most important aspects of the optimal performance of LED systems [1].

In the following graphs, it can be observed how important LED parameters vary with the junction temperature that are included in the data sheet provided by the manufacturer. Figure 25 shows how the luminous flux decays as the bonding temperature increases. For the junction temperature at the ambient temperature of 20 °C, which is 117 °C, the LEDs have a 20% decrease in luminous flux. By increasing the LED junction temperature, the power that is perceived as light by the human eye decreases.

Figure 26 shows the variation of the chromaticity coordinates with respect to the change in the junction temperature. For the junction temperature at the ambient temperature of 20 °C at which the LEDs of the Model are located (green line of Figure 26), the variation of the chromaticity coordinates is shown.

The variation of the junction temperature of the LED causes the chromaticity coordinates to change. We observed how the color quality of the light becomes worse. For the junction temperature corresponding to 20 °C ambient temperature, which is 117 °C, the variation of chromaticity varies causing a decrease in the color quality of the LED light, moving in the chromaticity diagram around 15%.

The theoretical and experimental data vary very slightly comparatively in Figure 27. In this way, we can verify that, from simulation techniques, we can obtain a very real approximation to the operating results of the Model without the need to build the prototype. The thermal dissipation simulation techniques allow to obtain very precise information, always linked to the precision of the geometry of the design and the materials, as fundamental for the results to be as accurate as possible.

At the time of favoring heat dissipation, the dissipater is the most important component to reduce the binding temperature. Most heatsinks are designed with fins to increase their contact surface with air and dissipate more heat [42]. It is necessary to improve the design of the components of the luminaire to facilitate the dissipation of heat and favor the passage of air through the heatsink of the luminaires to lower the temperature of electronic devices. To improve cooling, auxiliary ventilation systems could be incorporated, to improve the flow of air from the interior to the exterior.

A major problem with high power LED luminaires is the heat generated inside. The distribution of the LEDs and the effect of the number of LEDs lit affect the junction temperature and strongly participate in the degradation of the LED [43]. Most luminaires contain aluminum heatsinks as a solution. Currently, materials with new alloys and formulations for effective thermal dissipation are being investigated. For example, aluminum nitrate has been developed to be applied as a thin layer in the dissipaters, which has certain advantages over conventional dielectrics based on polymers or ceramic substrates, such as excellent thermal dissipation and low thermal resistance [44]. The issue of thermal dissipation is the order of the day; materials are being investigated to be used as heat sinks that promote greater durability of LEDs due to an improvement in thermal dissipation [45].

## 5. Conclusions

With this work, we intend to present an analytical methodology to calculate the thermal dissipation of LED luminaires, previously, in the design phase. The method was validated by comparing the results of the theoretical thermal simulations on the basis of a previously designed luminaire model with the experimental data obtained in the laboratory.

The proposed methodology allows the simulation to be carried out in the design phase of the LED luminaires, before launching into the construction phase of the prototype. This allows to analyze and experiment with the model in a virtual environment, reducing the time and cost requirements associated with the tests performed.

This study is aimed at the preliminary thermal analysis of LED luminaires, to verify the design and materials of the luminaires and to check the temperature of the LEDs and their impact in the face of the good functioning of light-emitting diodes. The study observes the influence of the junction temperature of the LEDs, as a critical issue of the design phase that can seriously affect the functionality of the luminaire and causes a decrease in the useful life and variation of the light properties of the LEDs, such as the decrease in luminous flux and change in chromaticity coordinates.

All electronic devices and circuits generate excess heat and require greater attention to avoid premature failures. LEDs convert 45% of the energy applied to light and the remaining 55% into heat, which must be dissipated using a design and suitable materials so that the durability and properties of the LEDs are not affected by the increase in temperature [17].

The thermal simulations give the engineer information about the temperature and air flow inside the equipment, allowing engineers to design cooling systems to optimize design and reduce energy consumption, weight, and cost, and verifying that there are no problems when the equipment is built. In this work, we used thermal simulation software applying computational fluid dynamics (CFD) techniques to predict the temperature and air flow of an electronic system [34].

A good design of the luminaire, where the circulation of air is favored for the benefit of the dissipation of the junction temperature of the LEDs, favors the useful life and improves the lighting efficiency and reliability of the LEDs, which allows them to have better lighting properties. Apart from the design, a good selection of materials for the components of the luminaire where the thermal conductivity is high favors the thermal dissipation of the luminaire.

Addressing and trying to mitigate the LED fixture overheating issues associated with poor heat dissipation is key, as they end up affecting light quality, chromaticity, and color temperature, directly impacting the lifespan of luminaires and the health of users by altering circadian rhythms. That is why we consider it essential to design lighting equipment that allows high thermal dissipation [46]. This work proposes a methodology for the scientific researcher that satisfies the future need to design luminaires with LED technology that are sustainable for the environment, safe for human health, reliable and durable, supported by the proven results obtained.

## Figures and Tables

**Figure 1 ijerph-19-00752-f001:**
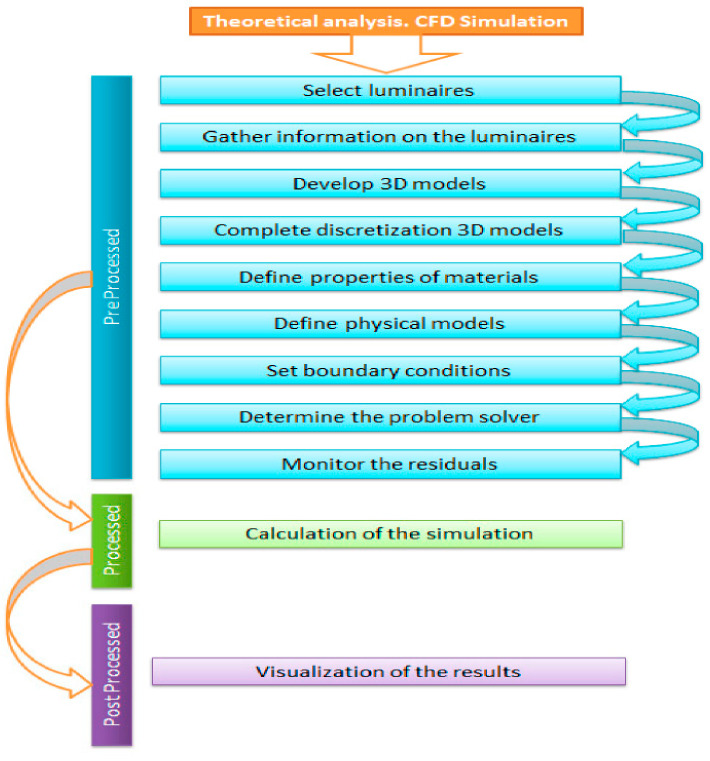
Flow diagram for the analysis of theoretical thermal dissipation. Source: our own elaboration.

**Figure 2 ijerph-19-00752-f002:**
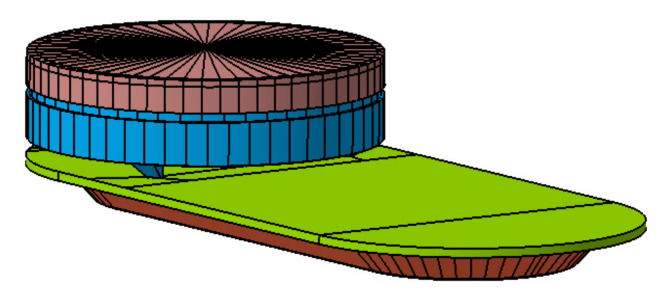
Modelling in 3D of the luminaire Model. Source: our own elaboration.

**Figure 3 ijerph-19-00752-f003:**
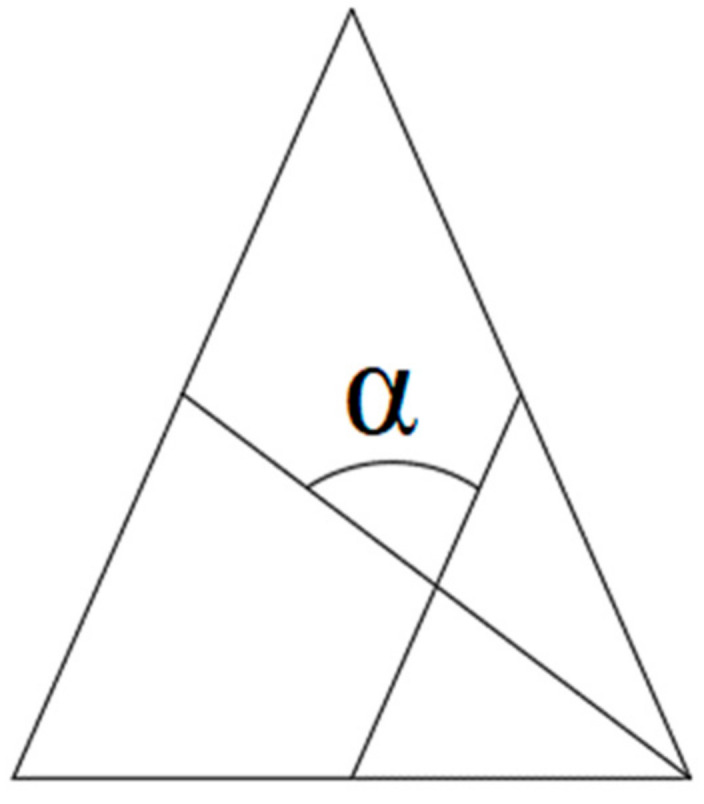
Definition of skew. Source: HyperMesh meshing control guide.

**Figure 4 ijerph-19-00752-f004:**
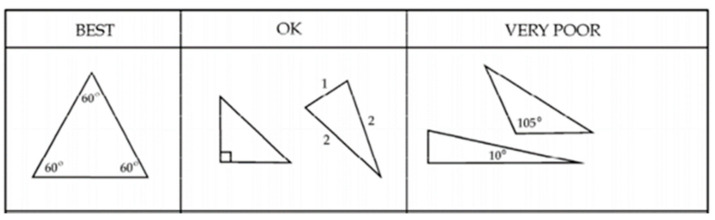
Mesh control. Source: HyperMesh meshing control guide.

**Figure 5 ijerph-19-00752-f005:**
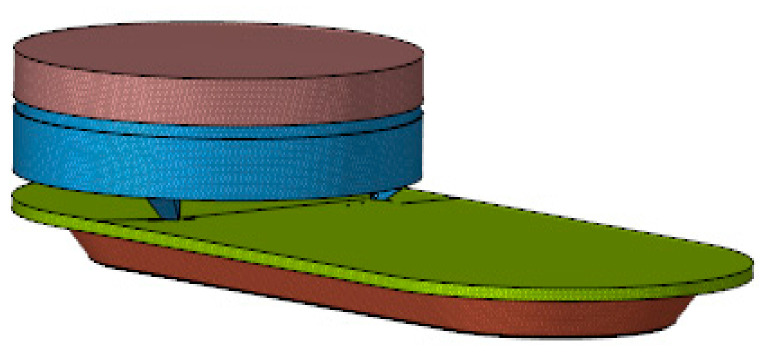
Discretized luminaire Model. Source: our own elaboration from the software HyperMesh.

**Figure 6 ijerph-19-00752-f006:**
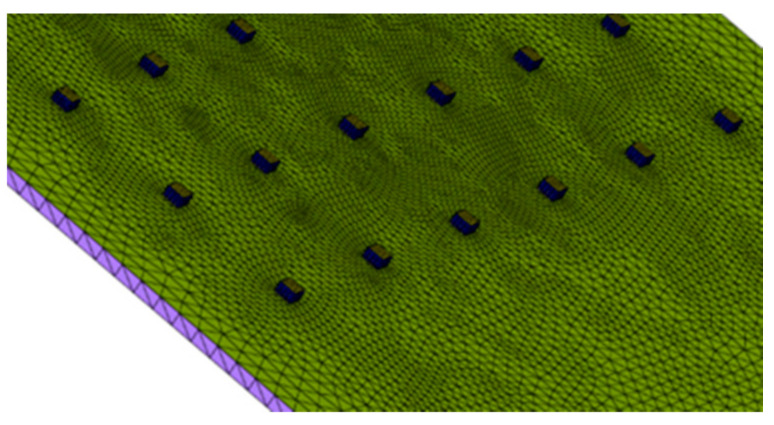
Discretized PCB and LED diode. Source: our own elaboration from the software HyperMesh.

**Figure 7 ijerph-19-00752-f007:**
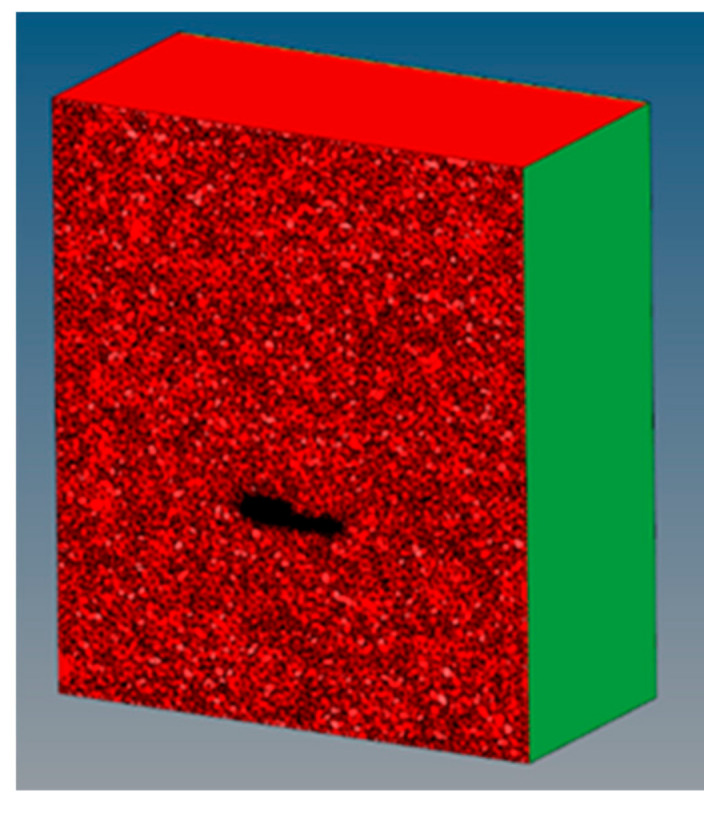
Average section of the components and air generated with tetrahedral Model. Source: our own elaboration from the software HyperMesh.

**Figure 8 ijerph-19-00752-f008:**
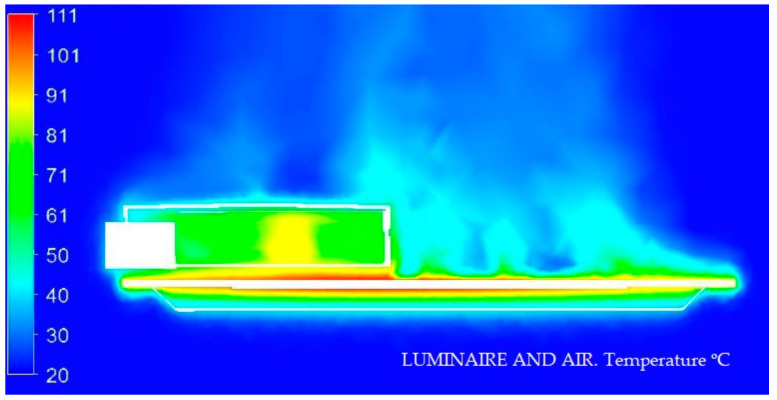
Temperature medium section of the luminaire at 20 °C. Source: HyperView software.

**Figure 9 ijerph-19-00752-f009:**
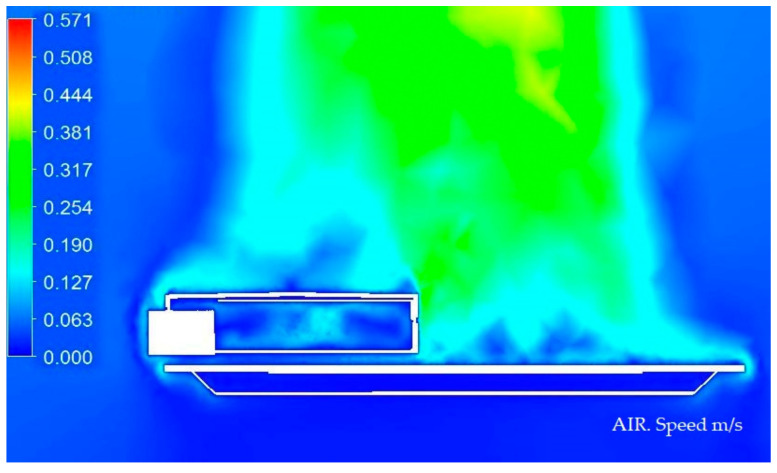
Speed section medium of the luminaire at 20 °C. Source: HyperView software.

**Figure 10 ijerph-19-00752-f010:**
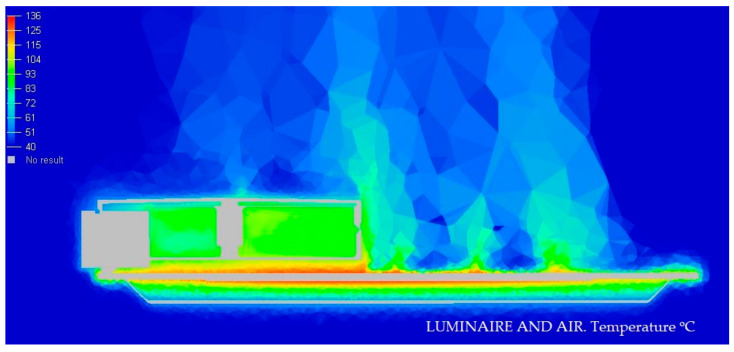
Temperature medium section of the luminaire at 40 °C. Source: HyperView software.

**Figure 11 ijerph-19-00752-f011:**
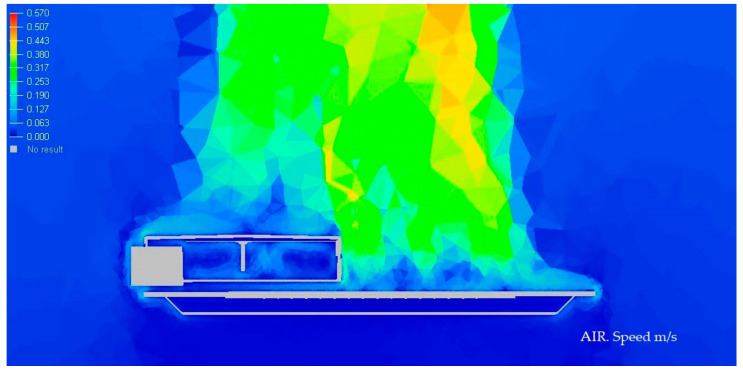
Speed section media of the luminaire at 40 °C. Source: HyperView software.

**Figure 12 ijerph-19-00752-f012:**
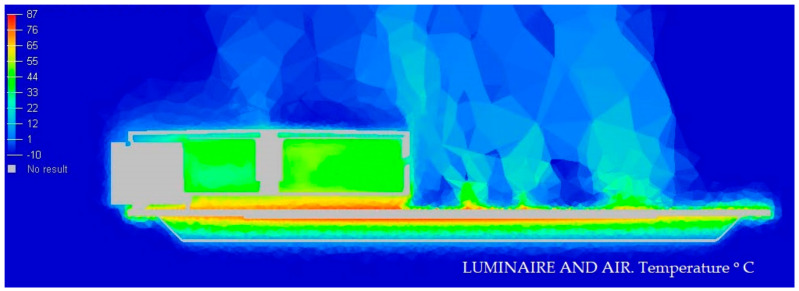
Temperature medium section of the luminaire at −10 °C. Source: HyperView software.

**Figure 13 ijerph-19-00752-f013:**
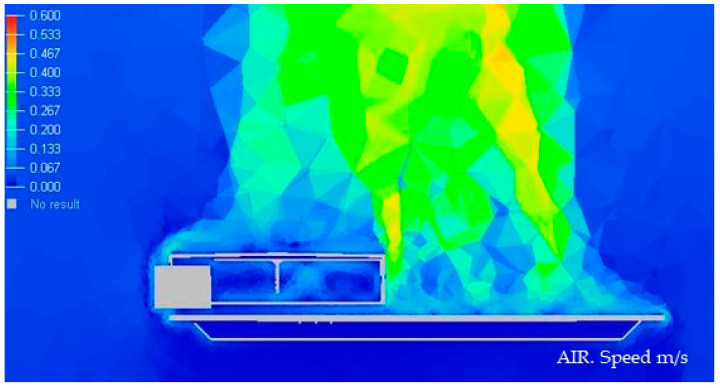
Speed section medium of the luminaire at −10 °C. Source: HyperView software.

**Figure 14 ijerph-19-00752-f014:**
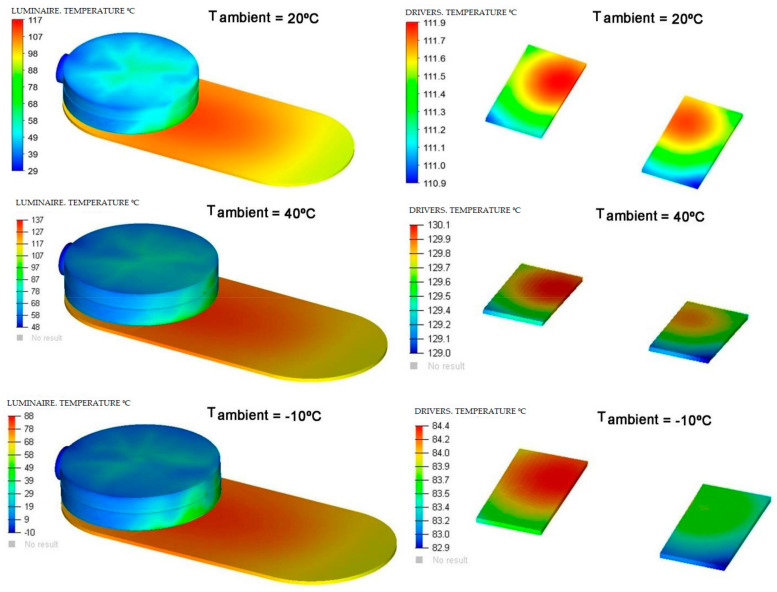
Temperature range of the complete luminaire and of the drivers base with respect to the ambient temperature in each simulation. Source: our own elaboration.

**Figure 15 ijerph-19-00752-f015:**
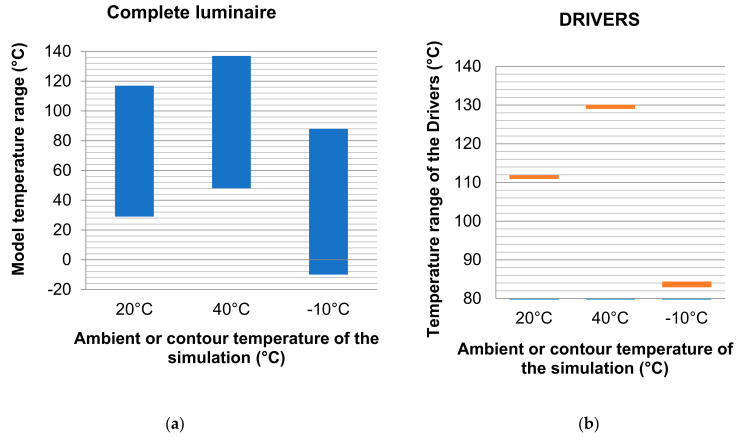
Temperature range of the complete luminaire (**a**) and of the drivers base (**b**) with respect to the ambient temperature in each simulation. Source: our own elaboration.

**Figure 16 ijerph-19-00752-f016:**
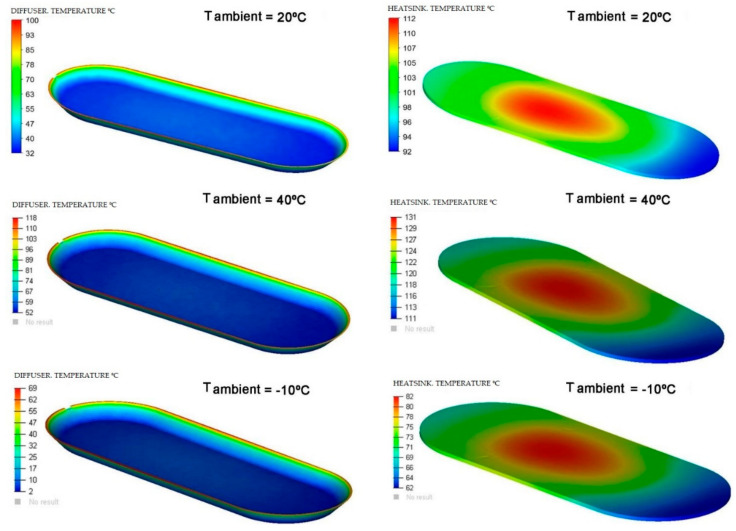
Comparison of dissipation of the diffuser (**left**) and of the heatsink (**right**) for different simulation temperatures. Source: our own elaboration from the software HyperView.

**Figure 17 ijerph-19-00752-f017:**
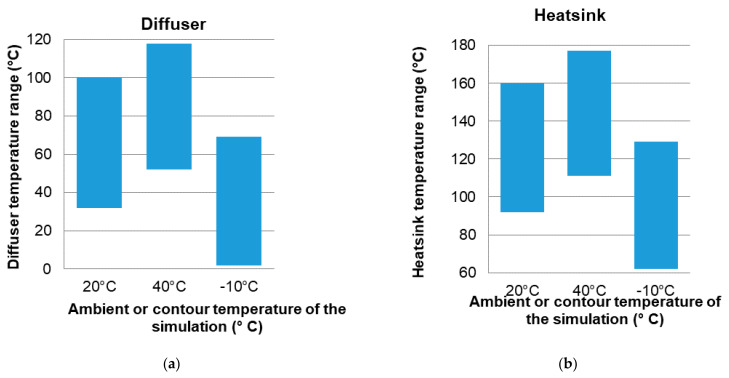
Temperature range of the diffuser (**a**) and the dissipater or heatsink (**b**) with respect to the ambient temperature in each simulation of the Model. Source: our own calculations.

**Figure 18 ijerph-19-00752-f018:**
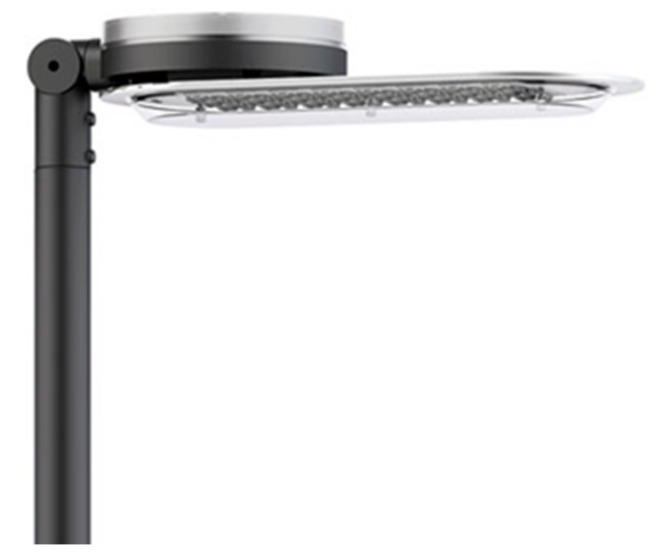
Luminaire Model. LED Air Series 7 of ATP Lighting. Source: ATP LED catalog lighting.

**Figure 19 ijerph-19-00752-f019:**
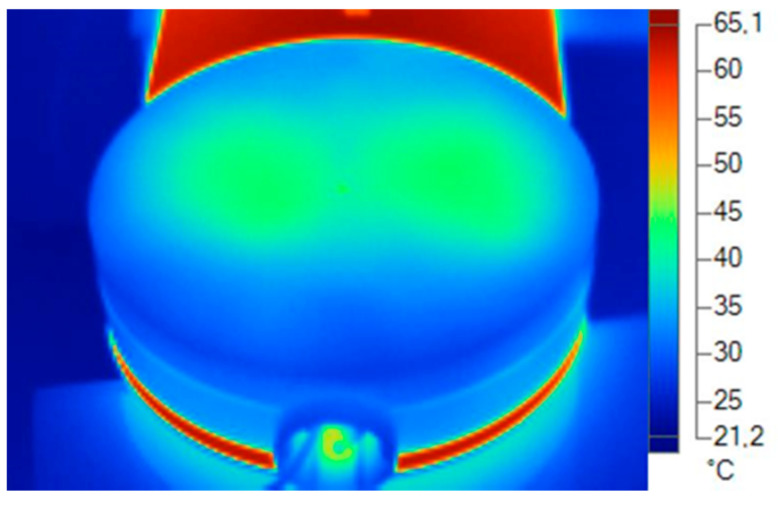
Thermal data of the cover. Blue and green area. Plastic emissivity 0.92. Source: Software SmartView 4.1 Fluke.

**Figure 20 ijerph-19-00752-f020:**
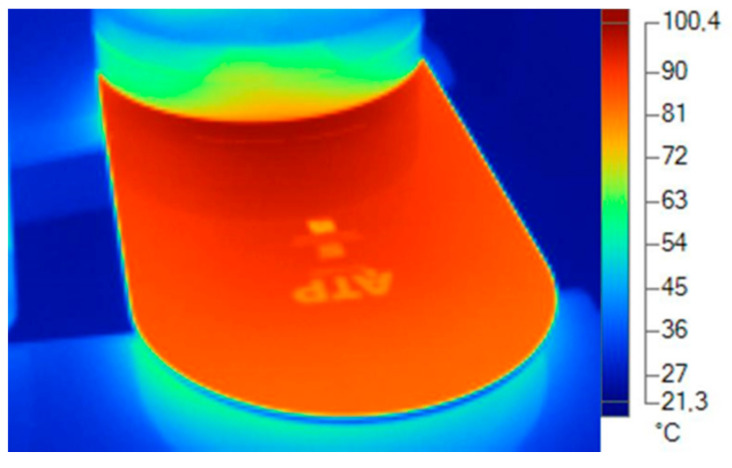
Thermal data of the heatsink. Reddish area: emissivity aluminum alloy 0.5. Source: Software SmartView 4.1 Fluke.

**Figure 21 ijerph-19-00752-f021:**
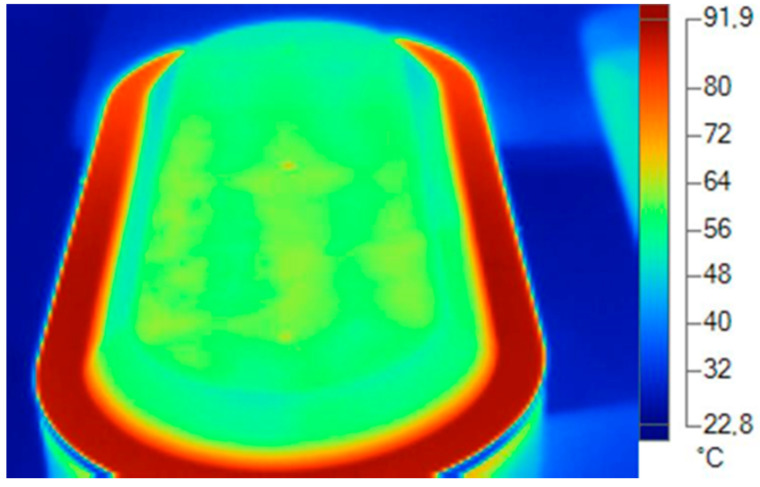
Thermal data of the heatsink. Reddish area: emissivity aluminum alloy 0.5. Source: Software SmartView 4.1 Fluke.

**Figure 22 ijerph-19-00752-f022:**
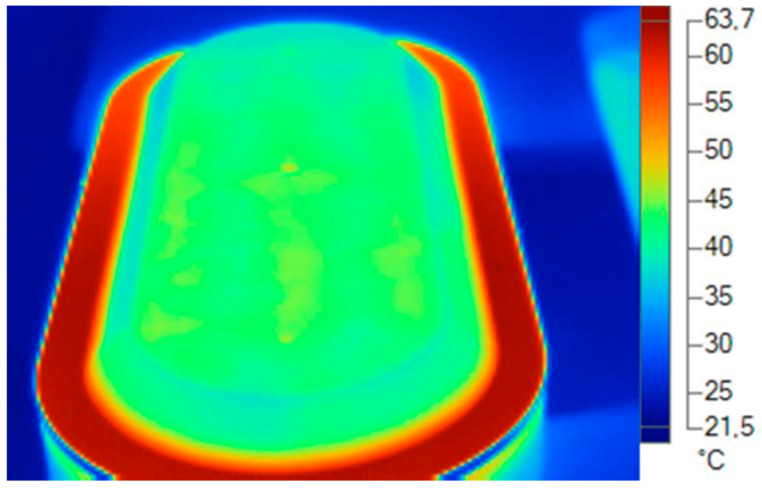
Thermal data of the diffuser. Green zone: plastic emissivity 0.92. Source: Software SmartView 4.1 Fluke.

**Figure 23 ijerph-19-00752-f023:**
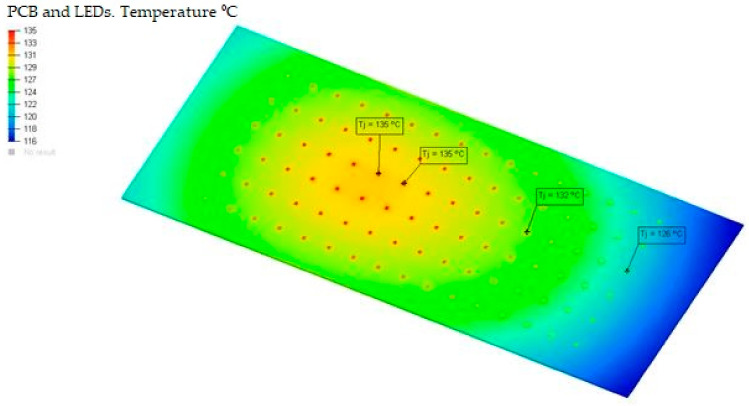
Summary of the joining temperatures of LEDs during the simulation at ambient temperature of 40 °C. Source: our own elaboration.

**Figure 24 ijerph-19-00752-f024:**
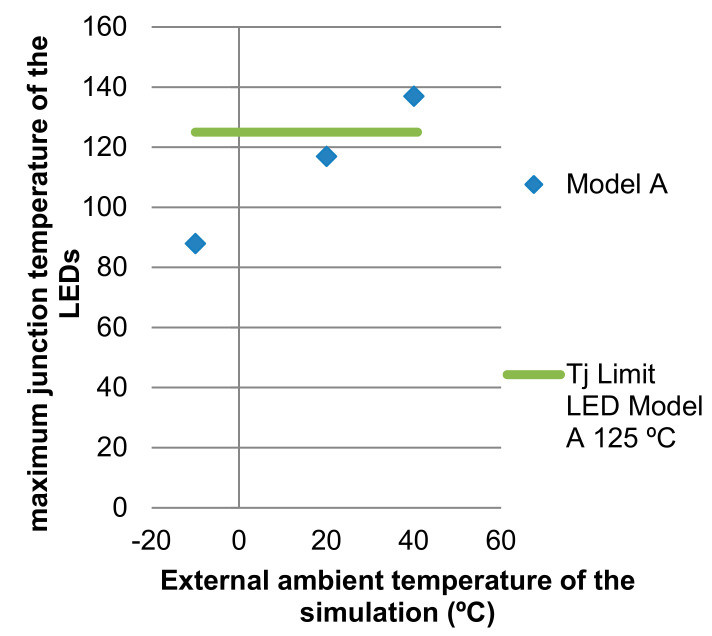
Variation of the maximum junction temperature of the LEDs with respect to the ambient temperature of the simulation, marking the limit junction temperature of each LED. Source: our own elaboration.

**Figure 25 ijerph-19-00752-f025:**
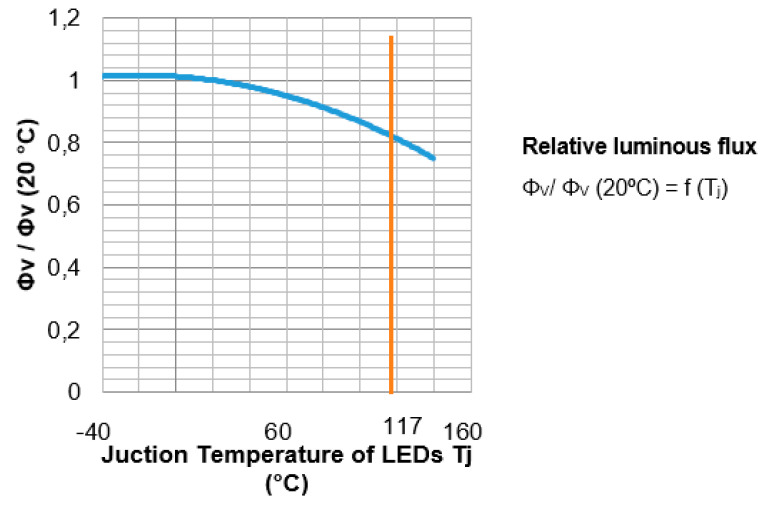
Junction temperature of LEDs (Tj) vs. luminous flux. Source: our own elaboration and data sheet LED Osram LUW CQAR (streetwhite).

**Figure 26 ijerph-19-00752-f026:**
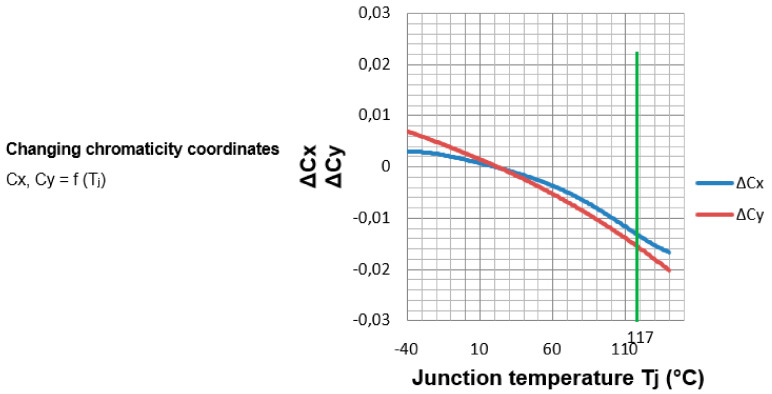
Junction temperature (Tj) vs. change of chromaticity coordinates. Source: our own elaboration and data sheet LED Osram LUW CQAR (streetwhite).

**Figure 27 ijerph-19-00752-f027:**
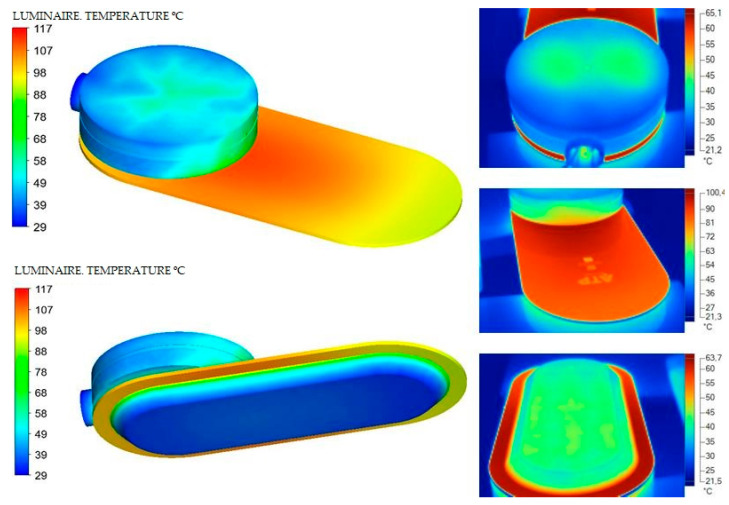
Comparative theoretical data (**left**) and experimental data (**right**). Source: our own elaboration from the software HyperView and SmartView 4.1.

**Table 1 ijerph-19-00752-t001:** Components, materials and powers. Source: ATP lighting S.A.

Components	Subcomponent	Part	Material	Information
Cover	-	-	PC	Opaque
Heatsink	-	-	Aluminum	Confidential
Equipment carrier	-	-	PA66–30FV	-
Equipment	-	-	PA66–30FV	-
Drivers		Microchips	Silicon	2 drivers; 6 W/driver
	PCB driver	Welding	Tin	
		Base	Aluminum	
	Housing	-	PC	
Diffuser	-	-	PC	Transparent
Chassis	-	-	PA66–30FV	-
PCB LED	-	-	Aluminum	-
LED	-	-	Copper/Material Rth	96 LED; 2 W/LED

**Table 2 ijerph-19-00752-t002:** Properties of the materials of the model. Source: commercial catalogs of the manufacturer’s Model (ATP).

Material	Density kg/m^3^	Specific Heat J/kg·K	Thermal Conductivity W/m·K
Aluminum	2750	961	200
Silicon	2330	700	148
Tin	7365	228	66.6
PA66–30FV	1370	2290	0.29
PC	1200	1250	0.19
Copper	8900	394	387
Material Rth	3300	780	52.91

The thermal resistance of the LED (Rth) of the luminaire of Model is 2.1 W/K.

**Table 3 ijerph-19-00752-t003:** Maximum temperature limit of the material versus temperature measured in the different parts of the Model at 20 °C. Source: our own elaboration.

Component	Material	Maximum Limit Temperature (°C)	Temperature Measured (°C)
Cover	PC	145	70
Heatsink	Aluminum	460	112
Equipment carrier	PA66–30FV	150	77
Equipment	PA66–30FV	150	112
Drivers Bases	Aluminum	460	112
Driver (Electronic)	Silicon (Weakest)	150	105
Diffuser	PC	145	100
Chassis	PA66–30FV	150	107
PCB	Aluminum	460	117

**Table 4 ijerph-19-00752-t004:** Maximum temperature limit of the material versus temperature measured in the different parts of the Model at 40 °C. Source: our own elaboration.

Component	Material	Maximum Limit Temperature (°C)	Temperature Measured (°C)
Cover	PC	145	90
Heatsink	Aluminum	460	131
Equipment carrier	PA66–30FV	150	97
Equipment	PA66–30FV	150	129
Drivers Bases	Aluminum	460	130
Driver (Electronic)	Silicon (Weakest)	150	130
Diffuser	PC	145	118
Chassis	PA66–30FV	150	123
PCB	Aluminum	460	137

**Table 5 ijerph-19-00752-t005:** Maximum temperature limit of the material versus temperature measured in the different parts of the Model at −10 °C. Source: our own elaboration.

Component	Material	Maximum Limit Temperature (°C)	Temperature Measured (°C)
Cover	PC	145	42
Heatsink	Aluminum	460	82
Equipment carrier	PA66–30FV	150	47
Equipment	PA66–30FV	150	83
Drivers Bases	Aluminum	460	84
Driver (Electronic)	Silicon (Weakest)	150	90
Diffuser	PC	145	69
Chassis	PA66–30FV	150	74
PCB	Aluminum	460	88

**Table 6 ijerph-19-00752-t006:** Luminaire power table. Source: manufacturer’s data.

Model	Nominal Power (W)	Number of LEDs	Power by LED (W/LED)	Power DRIVER (W)
ATP Aire Serie 7	204	96	2	12 (6 W/driver)

**Table 7 ijerph-19-00752-t007:** Driver features. Source: manufacturer’s data.

Model	Efficiency (%)	Nominal Power (W)	Operating Input Range (V)	Storage Temperature (°C)	Output Current (A)
MP4688	95	2–2.5	4.5–80	−65 to 150	Up to 1 A

**Table 8 ijerph-19-00752-t008:** Table of specifications of the thermal imager. Source: the user’s manual of the Ti 25 FLUKE thermal camera.

Attribute	Value
Thermal sensitivity	≤90 mK
Temperature Measurement Range	−20 → +350 °C
Maximum Accuracy of Temperature Measurement	±2 °C
Field of vision H × V	23 × 17°
Update frequency	9 Hz
Minimum Focus Distance	15 (Thermal Lens) cm, 46 (Visual Lens) cm
Type of Focus	Manual
Detector Resolution	160 × 120 pixel
Display size	3.7 plg
Display Resolution	640 × 480 pixel
Model number	Ti25

**Table 9 ijerph-19-00752-t009:** Junction temperatures of the LEDs obtained in the simulations (T_j_). Source: our own elaboration.

	External Ambient Temperature of the Simulation (°C)	Junction Temperature of the LEDs T_j_ (°C)
Temperature 1	40	135
Temperature 2	20	117
Temperature 3	−10	86

## Data Availability

We accept MDPI Research Data Policies Section at https://www.mdpi.com/ethics (accessed on 26 November 2021).

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
