# Peer review of "Impact of Thermal Dissipation on the Lighting Performance and Useful Life of LED Luminaires Applied to Urban Lighting: A Case Study"

_ijerph, 2022, doi:10.3390/ijerph19020752_

Round 1

Reviewer 1 Report

This manuscript “Impact of thermal dissipation on the lighting performance and  useful life of LED luminaires applied to urban lighting. Case  study” is a case study, that  has as goal to compare the CFD thermal results with the practical results of a thermal imaging camera.

Line 116 and next Table 1 Componet-driver Material-Aluminium Information- (two drivers 6W/driver). Please author read all the review and behind identify the elements of the driver/s and its functionality, there are need several thinks that clarify. Usually, a LED driver is a current injection control for LEDs, in some case with an additional AC-DC converter. If this correspond with the definition of your driver there are a gross mistake that the authors maintain in all manuscript. The aluminum is a material of the drivers, probably, but is not the main material and obviously is not the low limit for the low temperature limit material. A LED driver has electronic components and that is the weakness temperature element of a driver, and in several cases is the element that limit the lifetime of the luminaire.  In Table 363 the identify limit of driver is 460ºC when usually the temperature ambient limit for an electronic component is over 85ºC. Correct and discuss this in the manuscript. I show in the final part of the manuscript the considerations over the Tj LEDs , but I do not show the consideration of the electrical (power conversion) driver. Both thinks will be  considered from the beginning.

Related with the last paragraph, in Figure 14 the authors identify the driver with two separate plates, I do not understand your concept of driver. can you explain clearly?, because in Table 6 the driver has electronics parameters (6W) that do not correspond with two separate aluminium plates, or the lates figures. How is possible that the lower limit of temperature in table 4 and six will be the Cover and Diffuser. Have all the electronics components used in the luminaire -as white LEDs, and electronics components of driver- working limits over 145ºC?

The description of the luminaire is basically the description of a passive set of materials, and that is not this case. The parameters of electronics components used in the luminaire appear and the end, and in the case of driver are avoid, this suppose misunderstand the goal and the main relation with the luminaire lifetime.

There are other rough problem with the electronics terms of the manuscript in general and with the efficiency of the white LEDs in particular. If a do not has mistaken, this suppose that the authors will be correct or repeat the simulatios. The author speaks in line 44 “The LEDs disperse 15% of the  energy applied in light and the remaining 85% in heat. High-power chips increasingly  emit more heat, which leads to performance degradation” and line 554 “The LEDs convert 15% of the energy applied to light and the  remaining 85% into heat,”. This is efficiency for some LED technologies, but  is not a common percentage for white LEDs, where the objective is roughly  the 50%.  Furthermore, the authors write the identification of the LED and usually this suppose a direct or indirect identification of the real efficiency, in this case the information is clearly in the datasheet. Page 4 of document LUW CQAR (Streetwhite)_ EN.pdf on https://www.osram.com/ecat/OSLON%C2%AE%20Square%20LUW%20CQAR/com/en/class_pim_web_catalog_103489/prd_pim_device_2402488/

"Electrical" thermal resistance junction / solder point

"Elektrischer" Warmewiderstand Sperrschicht /

Lotpad

(with efficiency ηe = 45 %)

(typ.) Rth JS el 2.1 K/W

What is coherent witn the Rth information of the authors.

The text (with efficiency ηe = 45 %) identify a ratio illumination power/electrical power , and suppose in this case that the LEDs convert 45% of the energy/power applied to light and the  remaining 55% into heat. For a disambiguation of the parameter efficiency ηe applied in the omrom manufacturer you can consult page https://dammedia.osram.info/media/resource/hires/osram-dam-2496735/Package%20related%20thermal%20resistance%20of%20LEDs.pdf document “Package related thermal resistance of LEDs.pdf” page 3 of 9 and example of Tj calculus in page 7 of 9.

Additional considerations are:

In Table 2 it is need included the PCB (Printed Circuit Board) (alluminium) material that appeared at Table 1.

Line 223 en-ergy Line 260 seg-regated, This is common in the manuscript, please consult to the editor the correct mode for finish the lines, usually the words are whole. Cut the words is not a good solution, this problem could be caused to the conversion .doc to pdf of mdpi. Other related problem appear at Line 186 to 288, where there are problems with the font of the numbers.

Line 319 I think is best this redaction. …use temperature and “air” speed …

Line 321 “In the next in figures, we graphically display in colour the temperature of different 321 parts of model at 20°C”. (“in figures” I think is not a good expression)..

Figure 8. text in the figure is in Spanish. Use English. This problem is general in all temperature and air speed figures.

Table 6. The authors would explain the combination of a power driver of 12W for a nominal power luminaire of 204W.

Line 411, you can identify in contributions and acknowledges the origin of the thermal camara, but in the manuscript is not a common issue.

Some of the attributes of the camera are irrelevant, I think on weight, height, width and length. Only use the relevant information for the researchers.

In section 3.2 it is necessary identify the boundary real conditions, not only the stabilized 20ºC. Identify here that the measurements are in laboratory, that suppose probably 0 m/s air speed, and a climatized ambient.

In figure 21 and 22 there are a cover over the LEDs, additionally an image with that dismount for see the outside temperature of the LEDs will be interesting and relevant.

In general, could be an interesting paper, but not in the present condition. It need additional work for to be published.

Author Response

Response to Reviewer 1:

Comments and Suggestions for Authors

This manuscript “Impact of thermal dissipation on the lighting performance and useful life of LED luminaires applied to urban lighting. Case study” is a case study that has as goal to compare the CFD thermal results with the practical results of a thermal imaging camera.

Thank you very much for your comments.

We sincerely appreciate your comments and hope that the changes made to the document will be appreciated by the reviewer.

Then we will try to answer the questions made by the reviewer one by one. In addition we will introduce in the original manuscript the changes in red color

We take this opportunity to thank the reviewer for his interesting comments that we have included in the text.

General comments

Line 116 and next Table 1 Componet-driver Material-Aluminium Information- (two drivers 6W/driver). Please author read all the review and behind identify the elements of the driver/s and its functionality, there are need several thinks that clarify. Usually, a LED driver is a current injection control for LEDs, in some case with an additional AC-DC converter. If this correspond with the definition of your driver there are a gross mistake that the authors maintain in all manuscript. The aluminum is a material of the drivers, probably, but is not the main material and obviously is not the low limit for the low temperature limit material. A LED driver has electronic components and that is the weakness temperature element of a driver, and in several cases is the element that limit the lifetime of the luminaire.  In Table 363 the identify limit of driver is 460ºC when usually the temperature ambient limit for an electronic component is over 85ºC. Correct and discuss this in the manuscript. I show in the final part of the manuscript the considerations over the Tj LEDs , but I do not show the consideration of the electrical (power conversion) driver. Both thinks will be  considered from the beginning.

Thank you very much for your interesting comments. Indeed, we agree with the reviewer and we have clarified this correct consideration in the manuscript. Thank you.

The drivers used in the luminaire have two functions:

  • Transform alternating current (AC) into direct current (DC), which is used by LED for their correct operation (AC-DC Converter)
  • Adapt the output voltage and current to the LED requirements.

So, we have introduced in Table 1 the different elements that compound the drivers.

Page 4 Line 143.

Regarding the drivers, they are not composed of only one material, as it is a combination of several subcomponents. The weakest components to the temperature are the microchips that are assembled on the driver PCB, then in the comparison we have divided the driver information in two rows (bases and electronics components) and for electronics we will select the temperature limit of the microchips for the evaluation of the driver.

Page 11, Line 393, Tables 3, 4 and 5

In the Tables 3, 4 and 5 we have selected as a limit temperature for the driver (Electronic) 150ºC because the chip integrated mounted in the driver PCB for current injection control for the LEDs is MP4688, its datasheet shows that the storage temperature is 150ºC. Attached you can find this information.

The relevant electrical characteristics of the MP4688 driver are attached. The efficiency of power conversion is up to 95% with wide 4,5V to 80V operating input range and thermal shutdown protection temperature of 150ºC. It delivers a constant current of up to 1A to high power LEDs.

The nominal power of this driver is 2-2,5W (Typical value).

The field of application is industry and general lighting.

Page 15, Line 448.

Related with the last paragraph, in Figure 14 the authors identify the driver with two separate plates, I do not understand your concept of driver. can you explain clearly?, because in Table 6 the driver has electronics parameters (6W) that do not correspond with two separate aluminium plates, or the lates figures. How is possible that the lower limit of temperature in table 4 and six will be the Cover and Diffuser. Have all the electronics components used in the luminaire -as white LEDs, and electronics components of driver- working limits over 145ºC?

Thank you very much for your comments.

The concept of driver is:  There are two drivers that controls the luminaire, with the electrical and thermal characteristics we have explained in previous question. These two drivers are supported on two aluminium bases showed in Figure 14, so this temperature in Figure 14 refers to aluminium bases of the drivers.

In tables 3, 4 and 5 we have clarified this question dividing the drivers into plates and electronics components.

All electronics components of the drivers have working limit under 150ºC, as explained previously.

It was indeed a mistake. We have clarified the correct data in the table and in the text.

Page 11 Line 393

The description of the luminaire is basically the description of a passive set of materials, and that is not this case. The parameters of electronics components used in the luminaire appear and the end, and in the case of driver are avoid, this suppose misunderstand the goal and the main relation with the luminaire lifetime.

We again appreciate the comment to the reviewer. We have proceeded to clarify this question in the text,

We have introduced a new Table, named table 7, with the main electronics drivers parameters used for control the luminaire.

Page 15 Line 448.

There are other rough problem with the electronics terms of the manuscript in general and with the efficiency of the white LEDs in particular. If a do not has mistaken, this suppose that the authors will be correct or repeat the simulatios. The author speaks in line 44 “The LEDs disperse 15% of the  energy applied in light and the remaining 85% in heat. High-power chips increasingly  emit more heat, which leads to performance degradation” and line 554 “The LEDs convert 15% of the energy applied to light and the  remaining 85% into heat,”. This is efficiency for some LED technologies, but  is not a common percentage for white LEDs, where the objective is roughly  the 50%.  Furthermore, the authors write the identification of the LED and usually this suppose a direct or indirect identification of the real efficiency, in this case the information is clearly in the datasheet. Page 4 of document LUW CQAR (Streetwhite)_ EN.pdf on https://www.osram.com/ecat/OSLON%C2%AE%20Square%20LUW%20CQAR/com/en/class_pim_web_catalog_103489/prd_pim_device_2402488/

"Electrical" thermal resistance junction / solder point

"Elektrischer" Warmewiderstand Sperrschicht /

Lotpad

(with efficiency ηe = 45 %)

(typ.) Rth JS el 2.1 K/W

What is coherent witn the Rth information of the authors.

The text (with efficiency ηe = 45 %) identify a ratio illumination power/electrical power , and suppose in this case that the LEDs convert 45% of the energy/power applied to light and the  remaining 55% into heat. For a disambiguation of the parameter efficiency ηe applied in the omrom manufacturer you can consult page https://dammedia.osram.info/media/resource/hires/osram-dam-2496735/Package%20related%20thermal%20resistance%20of%20LEDs.pdf document “Package related thermal resistance of LEDs.pdf” page 3 of 9 and example of Tj calculus in page 7 of 9.

It was indeed a mistake. We have clarified the correct data in the text. Thank you.

Page 2 Line 62 and 63

Page 22 Line 593 and 594

Additional considerations are:

In Table 2 it is need included the PCB (Printed Circuit Board) (alluminium) material that appeared at Table 1.

Thank you very much for the comment

Aluminium was included in the first row of Table 2, however, it has been highlighted in red.

Also, in the same table, we have included some materials related to the driver. It can be found in red

Page 6 Line 216

Line 223 en-ergy Line 260 seg-regated, This is common in the manuscript, please consult to the editor the correct mode for finish the lines, usually the words are whole. Cut the words is not a good solution, this problem could be caused to the conversion .doc to pdf of mdpi. Other related problem appear at Line 186 to 288, where there are problems with the font of the numbers.

Thank you very much for the comment.  It seems to be a problem with the mdpi .doc template.

Line 319 I think is best this redaction. …use temperature and “air” speed …

We appreciate the reviewer's comment again. We have modified this redaction. Thank you very much.

Page 9 Line 356.

Page 10 Line 368.

Page 11 Line 380.

Line 321 “In the next in figures, we graphically display in colour the temperature of different 321 parts of model at 20°C”. (“in figures” I think is not a good expression)..

We proceed to change. Thank you

Page 9 Line 358.

Page 10 Line 370.

Page 11 Line 382

Figure 8. text in the figure is in Spanish. Use English. This problem is general in all temperature and air speed figures.

Thank you very much for the comment. We have carried out an in depth review of all the Spanish words in all figures of the manuscript. Thank you

Table 6. The authors would explain the combination of a power driver of 12W for a nominal power luminaire of 204W.

Thank you very much for the comment. We explain in the text.

As we have explained in previous questions, the drivers used for control LEDs is MP4688 high power control LEDs. The luminaire has two drivers PCB, and each driver PCB has three MP4688. Each integrated MP4688 controls 16 LEDs (16*3*2=96 LEDs in the luminaire)

Each MP4688 has nominal power of 2W (datasheet). As we have 2 W/driver* 3 drivers/PCB*2 PCB = 12 W

Page 15 Line 451.

Line 411, you can identify in contributions and acknowledges the origin of the thermal camara, but in the manuscript is not a common issue.

We have changed in the text. Thank you

Page 15 Line 457.

Some of the attributes of the camera are irrelevant, I think on weight, height, width and length. Only use the relevant information for the researchers.

We have removed some attributes of the camera in the table 8. Thank you

Page 16 Line 465.

In section 3.2 it is necessary identify the boundary real conditions, not only the stabilized 20ºC. Identify here that the measurements are in laboratory, that suppose probably 0 m/s air speed, and a climatized ambient.

Thanks. We proceed to clarify. Thank you

Page 15 Line 457.

In figure 21 and 22 there are a cover over the LEDs, additionally an image with that dismount for see the outside temperature of the LEDs will be interesting and relevant.

We did not take this images without cover with thermal imager camera during experimental results phase at laboratory.

In general, could be an interesting paper, but not in the present condition. It need additional work for to be published.

We take this opportunity to thank the reviewer for his interesting comments that we have included in the text.

Reviewer 2 Report

The purpose of the article titled »Impact of thermal dissipation on the lighting performance and useful life of LED luminaires applied to urban lighting. Case  study« is to compare the results previously obtained, at different contour tem-19 peratures, by theoretical thermal simulation of the 3D model of LED street lighting luminaire 20 through the ANSYS Fluent simulation software.

The article is interesting and appropriately structured. It is interesting for designers and constructors of street lamps. I do not see much emphasis on the fact that the article contributes to environmental research and public health, which is the main topic of the journal. I definitely suggest a textual addition in this direction in the chapters Abstract, Introduction and Conclusion.

I leave the final assessment of the suitability of the topic for publication in the International Journal of Environmental Research and Public Health to the editor.

My other comments are as follows:

Line 84:

Abbreviation CFD is not explained.

Line 203: The equation 4 is not derived. Where is it taken from? The reference is not specified. The naming of symbols  and  is incorrect. Name the symbols correctly.

Line 207-212: The list of symbols is not complete. Where is ?

Line 224: The equation 5 is not derived. Where is it taken from? The reference is not specified.

Line 237: T = local temperature in °K?

Line 244: The equation 6 is not derived. Where is it taken from? The reference is not specified.

Name the symbols »Nfaces« and »f«.

Line 286: .... residual levels of 10e-4 ....... the levels of 10e-5: Show where this information came from. Which graphics shows this?

Line 324 – 337: Figure 8-10: Describe the pictures. Explain what can be seen from these pictures. How do the pictures differ from each other?

Line 376: Figure 14: Is the sequence of pictures correct? I recommend a description of the differences between the pictures.

Figures 14 and 15 are not consistent. Why not?

Line 385: Figure 16: Is the sequence of pictures correct? I recommend a description of the differences between the pictures.

Figures 16 and 17 are not consistent. Why not?

Line 408: Table 6: Error in table

Line 416: precision not pressure

Figure 20 – 22: I recommend a description of the differences between the pictures.

Figure 27: I recommend a description of the comparative analysis between the simulation and the measured results.

Author Response

Response to Reviewer 2:

Comments and Suggestions for Authors

The purpose of the article titled »Impact of thermal dissipation on the lighting performance and useful life of LED luminaires applied to urban lighting. Case  study« is to compare the results previously obtained, at different contour tem-19 peratures, by theoretical thermal simulation of the 3D model of LED street lighting luminaire 20 through the ANSYS Fluent simulation software.

The article is interesting and appropriately structured. It is interesting for designers and constructors of street lamps. I do not see much emphasis on the fact that the article contributes to environmental research and public health, which is the main topic of the journal. I definitely suggest a textual addition in this direction in the chapters Abstract, Introduction and Conclusion.

I leave the final assessment of the suitability of the topic for publication in the International Journal of Environmental Research and Public Health to the editor.

Thank you very much for your comments.

Thank you very much for your very timely comments. In fact, we have included both in the Abstract, as in the Introduction, as well as in the Conclusions, a clear justification supported by a wide bibliographical compilation that justifies the direct relationship between the need to design luminaires with effective thermal dissipation criteria, given the impact that It has this phenomena (as we collect in the supporting bibliography) on the depreciation of the luminous intensity, the chromaticity and the useful life of the luminaires. This fact has a direct impact on human health and the alteration of the users' circadian rhythms and on the environmental impact of the shortening of the durability and useful life of the luminaires in the LCA (Life cycle analysis), carbon footprint and the environmental impacts of early replacement of LED luminaires due to anticipated deterioration

We sincerely appreciate your comments and hope that the changes made to the document will be appreciated by the reviewer.

Then we will try to answer the questions made by the reviewer one by one. In addition we will introduce in the original manuscript the changes in red colour

Abstract:

“There are many studies that show a direct relationship between the low quality of LED lighting, related to the aging of the equipment or the overheating of the same, observing the depreciation of the intensity of the light and the visual chromaticity performance that can reach to affect the health of users by altering circadian rhythms. On the other hand, the shortened useful life of the luminaires due to thermal stress has a direct impact on the LCA (life cycle analysis) and its environmental impact indirectly affecting human health.”

Introduction

“On the other hand, he increasing use of LED lights has modified the natural light environment dramatically, posing novel challenges to both humans and wildlife. Indeed, several biomedical studies have linked artificial light at night to the disruption of circadian rhythms[4], with important consequences for human health, such as the increasing occurrence of metabolic syndromes [5], cancer [6], and reduced immunity.[7]. In this sense, there are many studies that clearly show the adverse effects on the health of poor lighting, including light depreciation as a consequence of deterioration, faulty thermal dissipation, or the devaluation of the light intensity and chromaticity of the luminaire.[8][9]

The chromaticity, electrical properties and thermal properties of LED devices are highly dependent on each other[10], therefore, heat dissipation plays an important role in improving the efficiency and reliability of LED lighting[11].  “ Deficiencies in artificial lighting, loss of light intensity, or changes in chromaticity can have serious consequences on human health, in relation to circadian rhythms[12][13]. This is why it is extremely important to install energy-efficient outdoor lighting equipment but at the same time low impact on human health.[14][15].Shorter wavelengths of light preferentially disturb melatonin secretion and cause circadian phase shifts, even if the light is not bright.[13].”

The object of this study is to propose an analysis methodology to calculate the theoretical thermal dissipation, a priori, depending on the design, the materials and the operating ambient temperature of the LED luminaires to later compare and discuss the theoretical results with the experimental ones made with a thermographic camera. To do this, we will thermally analyse, according to the proposed method, a LED luminaire, carrying out simulations, at different working environment temperatures [24]. “Considering that thermal dissipation is essential to maintain the luminous properties of LED luminaires, directly impacting the useful life of the equipment and its light quality[25]; in direct relation to the impact that these deficiencies can have on the cir-cadian rhythms of people exposed to the light artificial defective.[26][27]. Likewise, the useful life of LED luminaires is related to the quality of the equipment and the ability of the materials to dissipate radiated heat, this aspect directly impacting the LCA (Life Cycle Analysis) on the carbon footprint [28] and environmental impacts of the manu-facture of new substitute equipment.[29][25]. Today it is very important to bet on sus-tainable luminaires with a low carbon footprint and high durability.[28]”

  1. Conclusions

“Addressing and trying to mitigate LED fixture overheating issues associated with poor heat dissipation is key as they end up affecting light quality, chromaticity, and color temperature. Directly impacting the lifespan of luminaires and the health of users by altering circadian rhythms. That is why we consider it essential to design lighting equipment that allows high thermal dissipation.[45]. This work proposes a methodolo-gy for the scientific researcher that satisfies the future need to design luminaires with LED technology that are sustainable for the environment, safe for human health, relia-ble and durable, supported by the proven results obtained.”

My other comments are as follows:

Line 84: Abbreviation CFD is not explained.

Thank you very much for your comments. We have explained it in the manuscript.

Page 3 Line 111

Line 203: The equation 4 is not derived. Where is it taken from? The reference is not specified. The naming of symbols  and  is incorrect. Name the symbols correctly.

Thank you very much for your comment. We have modified it in the manuscript.

Page 7 Line 241

Line 207-212: The list of symbols is not complete. Where is ?

Thank you very much for your comment. We have modified it in the manuscript.

Page 7 Line 250 and 251

Line 224: The equation 5 is not derived. Where is it taken from? The reference is not specified.

Thank you very much for your comment. We have modified it in the manuscript.

Page 7 Line 266

Line 237: T = local temperature in °K?

Thank you very much for your comment. We have modified it in the manuscript.

Page 8 Line 277

Line 244: The equation 6 is not derived. Where is it taken from? The reference is not specified.

Thank you very much for your comment. We have modified it in the manuscript.

Page 8 Line 284

Name the symbols »Nfaces« and »f«.

Thank you very much for your comment. We have modified it in the manuscript.

Page 8 Line 286 and Line 287

Line 286: .... residual levels of 10e-4 ....... the levels of 10e-5: Show where this information came from. Which graphics shows this?

Thank you very much for your comment.

Attached you can find a graphics regarding the level of residuals obtained after the simulations. Thanks

Line 324 – 337: Figure 8-10: Describe the pictures. Explain what can be seen from these pictures. How do the pictures differ from each other?

Thank you very much for your comment.

These figures show the results of the temperature reached in the luminaire and the air velocity around the luminaire with respect to the three simulated boundary temperatures.

Line 376: Figure 14: Is the sequence of pictures correct? I recommend a description of the differences between the pictures.

Thank you very much for your comment.

Figure 14 is a comparison between the results of the simulations in the three boundary conditions of the complete luminaire and the aluminium plates on which the 2 electronic drivers controlling the luminaire are mounted.

Page 13 Line 417 to 419

Figures 14 and 15 are not consistent. Why not?

Thank you very much for your comment.

Figure 15 is a summary of figure 14 in which it studies the temperature range that is reached both in the complete luminaire and at the base of the drivers under the three different boundary conditions.

Page 13 Line 417 to 419

Line 385: Figure 16: Is the sequence of pictures correct? I recommend a description of the differences between the pictures.

Thank you very much for your comment.

Figure 16 is a comparison between the results of the simulations in the three boundary conditions of the diffuser and the heatsink mounted in the luminaire.

Figures 16 and 17 are not consistent. Why not?

Thank you very much for your comment.

Figure 17 is a summary of figure 16 in which it studies the temperature range that is reached both in the diffuser and at the heatsink under the three different boundary conditions.

Line 408: Table 6: Error in table

Thank you very much for your comment.

We have reviewed the Table 6, there were some missing information we have added. Did you mean that. We have added Table 7

Page 15 Line 453

Line 416: precision not pressure

Thank you very much for your comment. We have modified it in the manuscript.

Page 15 Line 466

Figure 20 – 22: I recommend a description of the differences between the pictures.

Thank you very much for your comment.

These figures shows the experimental results regarding temperature reached in complete luminaire at 20ºC in the laboratory. We have modified the emissivities according the material and part of luminaire studied.

Page 16 Line 473 to 487

Figure 27: I recommend a description of the comparative analysis between the simulation and the measured results.

Thank you very much for your comment.

In the Figure 27, the theoretical and experimental data vary very slightly comparatively, in this way we can verify that from simulation techniques we can obtain a very real approximation to the operating results of the model without the need to build the prototype. The thermal dissipation simulation techniques allow to obtain very precise information, always linked to the precision of the geometry of the design and the materials, as fundamental for the results to be as accurate as possible.

Page 20 Line 557 to 562

We take this opportunity to thank the reviewer for his interesting comments that we have included in the text.

Reviewer 3 Report

This paper presented an analytical methodology to calculate the thermal dissipation of LED luminaires in the design phase. The results of the theoretical thermal simulations and the experimental data were provided.

Theoretically, the methodology is feasible. However, the procedure of modelling, and how to obtain the parameters of the model are not explained clearly. The simulation results and the experimental results are not fitted well.

Also, there are some errors in the reference list, e.g., the authors in Ref[4] are not expressed correctly.

 I suggest that the author carefully revise the paper and resubmit it for reconsideration.

Author Response

Response to Reviewer 3:

Comments and Suggestions for Authors

This paper presented an analytical methodology to calculate the thermal dissipation of LED luminaires in the design phase. The results of the theoretical thermal simulations and the experimental data were provided.

Thank you very much for your comments.

We sincerely appreciate your comments and hope that the changes made to the document will be appreciated by the reviewer.

Then we will try to answer the questions made by the reviewer one by one. In addition we will introduce in the original manuscript the changes in red color

We take this opportunity to thank the reviewer for his interesting comments that we have included in the text.

General comments

Theoretically, the methodology is feasible. However, the procedure of modelling, and how to obtain the parameters of the model are not explained clearly. The simulation results and the experimental results are not fitted well.

Thank you very much for your interesting comments.

We have carried out a thorough revision of the manuscript and have modified the abstract and the introduction where more than 10 bibliographic references have been added. Regarding the results section, we have explained in more detail the operation of the drivers that control the luminaire and we have also added a new table, called Table 7, with the main characteristics of the drivers.

Figure 27 shows a comparison of the theoretical and practical results in which it can be seen that they are quite close, although there are small differences due to the fact that the components are real and in the simulation they are considered ideal.

Finally, new findings have been added on the effects of poor heat dissipation on the health of users, for example affecting circadian rhythms.

Also, there are some errors in the reference list, e.g., the authors in Ref[4] are not expressed correctly.

We again appreciate the comment to the reviewer.

We have expressed correctly the reference of the author. Now renamed as reference 10

Page 23 Line 670

 I suggest that the author carefully revise the paper and resubmit it for reconsideration.

Thank you very much for your review work. We have proceeded to carry out an in depth review of our article

We take this opportunity to thank the reviewer for his interesting comments that we have included in the text.

Submission Date

27 November 2021

Date of this review

23 Dec 2021 03:27:43

Reviewer 4 Report

The given article treats an actual problem - the impact of thermal dissipation on the lighting performance and useful life of LEDs. 
The title of the paper and the abstract of the article are informative. However, based on the information in the abstract, it isn't easy enough for readers to understand the novelty of this study. The reference list consists of 29  papers that are properly referenced. The references are relevant. In the introduction section, the authors explained the background of the topic from different points of view; at the end of the introduction section, they did not outline the problem gap. In the discussion section, the authors discussed the results from multiple angles and placed into context. 

The points in the article which needs clarification, refinement, reanalysis, rewrites and/or additional information and suggestions for what could be done to improve the article: 

  1. It’s suggested to clarify the research gap at the end of the introduction section.
  2. The authors need to revise the conclusions,  clarify the impact of the proposed results on the useful life of LED luminaires. 
  3. It's better to indicate the references for the equations (4)-(6). 
  4. It's recommended to clarify the novelty of the study.

Author Response

Response to Reviewer 4:

Comments and Suggestions for Authors

The given article treats an actual problem - the impact of thermal dissipation on the lighting performance and useful life of LEDs. 

The title of the paper and the abstract of the article are informative. However, based on the information in the abstract, it isn't easy enough for readers to understand the novelty of this study. The reference list consists of 29  papers that are properly referenced. The references are relevant. In the introduction section, the authors explained the background of the topic from different points of view; at the end of the introduction section, they did not outline the problem gap. In the discussion section, the authors discussed the results from multiple angles and placed into context. 

Thank you very much for your comments.

We sincerely appreciate your comments and hope that the changes made to the document will be appreciated by the reviewer.

Then we will try to answer the questions made by the reviewer one by one. In addition we will introduce in the original manuscript the changes in red colour

We take this opportunity to thank the reviewer for his interesting comments that we have included in the text.

General comments

The points in the article which needs clarification, refinement, reanalysis, rewrites and/or additional information and suggestions for what could be done to improve the article: 

  1. It’s suggested to clarify the research gap at the end of the introduction section.

Thank you very much for your interesting comments.  

We have clarified the research gap at the end of the introduction section. Thank you.

We have highlighted in red the object of our study “The object of this study is to propose an analysis methodology to calculate the theoretical thermal dissipation, a priori, depending on the design, the materials and the operating ambient temperature of the LED luminaires to later compare and discuss the theoretical results with the experimental ones made with a thermographic camera.” And also we have added which parameters will allow to study the lighting performance and useful life of luminaires by comparing theoretical and practical results. “The comparison of results between the simulations and the real dissipation in the laboratory will allow to analyse the junction temperature of the LEDs because it is a parameter that cannot be measured through a thermographic camera.”

Page 2 Line 80.

  1. The authors need to revise the conclusions, clarify the impact of the proposed results on the useful life of LED luminaires. 

We again appreciate the comment to the reviewer. We have proceeded to clarify this question in the conclusions. We have reviewed de impact of the results on the useful life of the LED.

“This study is aimed at the preliminary thermal analysis of LED luminaires, to veriify the design and materials of the luminaires and to check the temperature of the LEDs and their impact in the face of the good functioning of the light-emitting diodes. Observing the influence of the junction temperature of the LEDs, as a critical issue of the design phase that can seriously affect the functionality of the luminaire. Causing a decrease in the useful life and a variation of the light properties of the LEDs like de-crease in luminous flux and change in chromaticity coordinates.” (Principal conclusion)”

Page 22 Line 593

  1. It's better to indicate the references for the equations (4)-(6). 

Thank you very much for your comments. We have added the references associates for the equations (4)-(6) and also we have renamed these questions due to a mistake. Equations (1), (2) and (3)

Page 7 Line 240

Page 7 Line 265

Page 8 Line 283

  1. It's recommended to clarify the novelty of the study.

Thank you very much for your review work.  We have clarified its novelty in the Conclusions section.

“This study allows the simulation to be carried out in the design phase of the LED luminaires, before launching into the construction phase of the prototype. Allowing to analyse and experiment with the model in a virtual environment, reducing the time and cost requirements associated with the tests performed.”

Page 21 Line 589

We take this opportunity to thank the reviewer for his interesting comments that we have included in the text.

Submission Date

27 November 2021

Date of this review

20 Dec 2021 21:29:17

Round 2

Reviewer 1 Report

The manuscript has been significantly improved.

The authors will consider a small range of the Y axis.  0-140ºC in left side or figure 15. Current range 140-230ºC. I do not show the significance but could be related with the maximum range of materials. Review this aspect in all graphics to homogenize the axes if possible., 

Author Response

Comments and Suggestions for Authors

The manuscript has been significantly improved.

The authors will consider a small range of the Y axis.  0-140ºC in left side or figure 15. Current range 140-230ºC. I do not show the significance but could be related with the maximum range of materials. Review this aspect in all graphics to homogenize the axes if possible.

Thank you very much for your interesting comment.

We have modified and homogenised the temperature ranges on the Y-axis of the graphs in figure 15 and 17 where it is shown that the maximum temperature of the components are within the operating temperature range.

Page 13 Line 423 (Figure 15)

Page 14 Line 432 (Figure 17)
